# Adipose-Derived Stem-Cell-Membrane-Coated PLGA-PEI Nanoparticles Promote Wound Healing via Efficient Delivery of miR-21

**DOI:** 10.3390/pharmaceutics16091113

**Published:** 2024-08-23

**Authors:** Huiyu Peng, Fangzhou Du, Jingwen Wang, Yue Wu, Qian Wei, Aoying Chen, Yuhan Duan, Shuaiguang Shi, Jingzhong Zhang, Shuang Yu

**Affiliations:** 1School of Biomedical Engineering (Suzhou), Division of Life Sciences and Medicine, University of Science and Technology of China, Hefei 230026, China; huiyupeng@mail.ustc.edu.cn (H.P.); qianwei@mail.ustc.edu.cn (Q.W.); cay0503@mail.ustc.edu.cn (A.C.); dyh0624@mail.ustc.edu.cn (Y.D.); sgshi@mail.ustc.edu.cn (S.S.); 2Suzhou Institute of Biomedical Engineering and Technology, Chinese Academy of Sciences, Suzhou 215163, China; du_fangzhou@163.com (F.D.); wangjw@sibet.ac.cn (J.W.); wuy@sibet.ac.cn (Y.W.); 3School of Medical Imaging, Xuzhou Medical University, Xuzhou 221004, China

**Keywords:** biomimetic nanoparticles, microRNA delivery, PLGA-PEI, stem cell, wound repair

## Abstract

miRNAs have been shown to be involved in the regulation of a variety of physiological and pathological processes, but their use in the treatment of diseases is still limited due to their instability. Biomimetic nanomaterials combine nanomaterials with cellular components that are readily modifiable and biocompatible, making them an emerging miRNA delivery vehicle. In this study, adipose-derived MSC membranes were wrapped around PLGA-PEI loaded with miR-21 through co-extrusion and later transplanted into C57BL/6 mice wounds. The wound-healing rate, epithelialization, angiogenesis, and collagen deposition were assessed after treatment and corroborated in vitro. Our study demonstrated that m/NP/miR-21 can promote wound healing in terms of epithelialization, dermal reconstruction, and neovascularization, and it can regulate the corresponding functions of keratinocytes, fibroblasts, and vascular endothelial cells. m/NP/miR-21 can inhibit the expression of PTEN, a gene downstream of miR-21, and increase the phosphorylation activation of AKT, which can then regulate the functions of fibroblasts. In conclusion, this provides a new approach to therapy for skin wounds using microRNA transporters and biomimetic nanoparticles.

## 1. Introduction

The intricate process of wound healing involves a myriad of cellular events, with microRNAs (miRNAs) emerging as pivotal regulators [1]. Highly expressed miR-21 in adipose mesenchymal stem cells (ADSCs) and their derivatives can target and inhibit the expression of phosphate and tension homology deleted on chromosome ten (PTEN) genes to function in the proliferative phases of skin repair and accelerate the healing process of wounds [2]. However, due to the instability of miRNAs themselves, they are highly susceptible to degradation by RNA enzymes [3,4], and poor intracellular delivery and aggregation within the endosomes of unmodified miRNAs result in inefficient regulation [5]. Therefore, there is an urgent need for efficient and safe delivery of miRNAs.

Over the last two decades, nanomedicine has rapidly grown [6]. With their ability to improve control over the localization and release of therapeutic agents, nanocarriers offer a safer and more effective approach for addressing a wide range of diseases [7]. Polymers have been used for drug delivery and have advantages such as high stability, easy uptake into cells by endocytosis, and targeting abilities for specific tissues or organs through adsorption or coating with ligand materials at the surface of the particles [8]. Among them, PLGA-PEI is a well-designed nanoparticle for nucleic acid delivery [9,10]. Many studies have shown that PLGA-PEI nanoparticles (NPs) can effectively enter cells through specific and non-specific endocytosis [11]. Intracellular retention of PLGA NPs slowly releases the encapsulated drug, resulting in sustained drug efficacy [12,13]. The PEI on the PLGA-PEI surface is water-soluble, linear, or branched polymers with a protonatable amino group [14,15], which has a high cationic charge density at physiological pH and is capable of forming non-covalent complexes with DNA, siRNA, and antisense oligonucleotides. Therefore, PLGA-PEI can carry large numbers of nucleotides through electrostatic interactions [16,17] and occupies an important position among polycationic polymers for gene delivery. However, the use of PEIs can simultaneously increase the cytotoxicity of NPs [14,18,19], thus requiring surface modification.

Cell membrane coating is an emerging technology for the surface modification of nanoparticles [20]. Cell membrane modification improves the biocompatibility and stability of nanoparticles [21] and enables long-distance cargo transportation in the bloodstream or allows active targeting of the drug delivery process [22,23]. Mesenchymal stem cells (MSCs) are multipotent progenitor cells with capabilities for both multi-lineage differentiation and self-renewal [24]. MSCs are unique in that their surfaces are rich in receptor ligands, which are extensively involved in cellular interactions and cell–matrix adhesion [25] and play a key role in the homing and migration of MSCs [26,27]. In addition, MSC membranes camouflaged with NPs minimize blood clearance of circulating therapeutic agents by the reticuloendothelial system (RES) and increase cell-specific uptake. Although many functions in MSCs do not transfer with the cell membrane, many passive interactions between cell surface markers and target tissues are preserved [28]. Therefore, the use of MSCs for surface modification of functional NPs to design biomimetic drug delivery systems has progressed in many fields [29].

Herein, we fabricated a mADSC-membrane-modified PLGA-PEI nanocarrier named m/NP/miR-21 as an efficient loading–delivery system for miRNA and validated its function in wound repair. m/NP/miR-21 has a typical shell–nucleus morphological structure, which makes it able to load miRNAs efficiently. The prepared particles are homogeneous in size, highly stable, biologically safe, and easily taken up by cells. m/NP/miR-21 can promote wound healing from various perspectives, such as epithelialization, neovascularization, and dermal reconstruction, and it can regulate the corresponding functions, such as keratinocyte migration and tube formation of vascular endothelial cells, in addition to increasing the level of phosphorylated activation of Akt by inhibiting the expression of PTEN, a gene downstream of miR-21, and, thus, regulating fibroblast migration, proliferation, and collagen formation.

## 2. Materials and Methods

### 2.1. Cell Culture

All studies involving human samples were conducted in accordance with the Ethical Guiding Principles for Human Embryonic Stem Cell Research and the Declaration of Helsinki. The donors of umbilical cord and skin samples signed informed consent forms, and ethical approval was obtained from the First Affiliated Hospital of Soochow University (2019-136).

mADSCs were isolated by digesting small pieces of adipose tissue with collagenase (SERVA, Catoosa, OK, USA; S1745401) and cultured in DMEM high-glucose medium (D/H; Sbjbio, Nanjing, China; BC-M-002) supplemented with 10% fetal bovine serum (FBS; ExCell, Suzhou, China; FDN500) [30].

Human umbilical vein endothelial cells (HUVECs) were isolated as previously described [31]. Briefly, collagenase was injected into the umbilical vein, and the digest was subsequently collected and centrifuged to obtain cells. An endothelial culture medium and additive kit (ECM; Sciencell, Carlsbad, CA, USA; 1001) were used as a medium for HUVECs.

For human fibroblast (hDF) preparation, the skin samples were incubated with 2 mg/mL dispase (Roche, Shanghai, China; 04942078001) overnight at 4 °C to separate the epidermis and dermis. The minced dermis was further processed and cultured as described in [32].

Human immortalized keratinocytes (Hacats) were purchased from Cell Bank/Stem Cell Bank, Chinese Academy of Sciences and cultured in DMEM/F12 (D/F; Sbjcell, Nanjing, China; BC-M-002) with 10% FBS.

### 2.2. Preparation of Nanocarriers

mADSC membranes were extracted according to the method of Yao et al. [33]. Briefly, cells were collected through digestion, washed with PBS (Sbjcell, Nanjing, China; BC-PBS-01), and subsequently resuspended in hypotonic buffer (1 mM NaHCO_3_, 0.2 mM EDTA, 1 mM PMSF; Sinopharm, Beijing, China) that was prechilled at 4 °C. The cell suspension was homogenized 20 times on ice in a handheld DOSS homogenizer (Lingrui, Xinyang, China) and then centrifuged at 3200× *g* for 5 min at 4 °C to remove unbroken cells and non-membrane fractions. The supernatant was centrifuged again at 15,000× *g* for 30 min at 4 °C to collect the cell membranes. The collected cell membranes were dispersed in PBS and stored at −20 °C for use after the determination of the cell membrane protein concentration using the BCA method.

For the preparation of the biomimetic miRNA nanocarriers (m/NP/miR), miRNA mimics (Genewiz, Suzhou, China) were mixed with PLGA-PEI nanoparticles (SunLipo, Shanghai, China) at a ratio of 1:50 (*w*:*w*) and then incubated in a water bath at 37 °C for 30 min. PLGA-PEI nanocarriers loaded with miRNA (NP/miR) were mixed with cell membranes at a ratio of 1:1 (*w*:*w*), and the mixture was extruded 7 times using a manual extruder (AVESTIN, Ottawa, ON, Canada; LF-1) and left to stand at room temperature for 2 min. Centrifugation was carried out for 1 h at 4000× *g* using a 50 kD ultrafiltration tube (Millipore, Burlington, MA, USA; UFC910024) to remove unloaded miRNAs. For membrane-coated miRNA (m/miR), microRNAs were mixed with membranes and co-extruded, followed by ultrafiltration. Membrane nanovesicles (mNVs) were extruded directly from cell membranes without any components.

### 2.3. Characterization of Nanocarriers

A Nano-Sight LM10 instrument (Malvern Instruments, Malvern, UK) was used to detect the average size and the zeta potential of prepared nanocarriers in each sample based on dynamic light scattering (DLS). All experiments were performed at RT and repeated three times.

The size distribution of different nanocarriers was measured using nanoparticle tracking analysis (NTA) with a NanoSight NS500 instrument (Malvern Instruments, Malvern, UK). Then, the nanocarrier size was analyzed using the NTA software version 3.0 (Malvern).

The morphology and size distribution of the nanocarriers were observed using transmission electron microscopy (TEM). Images were taken using a JEOL 1200EX transmission electron microscope (JEOL, Tokyo, Japan) operated at 80 kV.

The miRNA encapsulation rate was calculated as follows: Encapsulation Rate = (M0 − M1)/M0, where M0 is the total mass of miRNA, and M1 is the mass of free miRNA obtained after ultrafiltration.

The prepared m/miR-NC and m/NP/miR-NC were co-incubated with 10% FBS at room temperature, and samples were taken after 0, 2, 4, 6, and 8 h of incubation, followed by nucleic acid gel electrophoresis for detecting the release and degradation of miRNAs in the biomimetic nanocarriers in a serum environment.

### 2.4. Cell Uptake

PKH26 (MKbio, Shanghai, China; MX4021) was employed to label the membrane coatings according to the manufacturer’s instructions. PKH26-labeled nanocarriers were then incubated with different types of cells at 37 °C for 6 h. Cells were then fixed in 4% paraformaldehyde (Sigma-Aldrich, St. Louis, MO, USA; 158127) for 15 min and blocked with 1% FBS for 20 min. The nuclei were stained with 4′,6-diamidino-2-phenylindole (DAPI; Beyotime, Shanghai, China; C1002) for 20 min at room temperature.

Samples were observed through confocal microscopy (Nikon, Tokyo, Japan; A1R HD25), and the mean fluorescence intensity of PKH26 was measured for each group at the measurement time points.

### 2.5. Scratch Wound Assay

The hDFs and Hacats were cultured at a density of 1 × 105 cells per well in 24-well plates and incubated until confluence. Monolayers of cells were scratched using 200 μL tips and washed with PBS to remove detached cells. Cells were then cultured in serum-free medium (D/H for hDFs and D/F for HUVECs) with a negative control (PBS), miR-21 (25 nM), m/miR-21 (25 nM), and m/NP/miR-21 (25 nM). Samples were imaged after 0, 24, and 48 h. The wound closure area was calculated as follows: Scratch Area (%) = An/A0 × 100%, where A0 denotes the initial scratch area, and An denotes the remaining area of the wound at the measurement point. The experiment was replicated three times, and the scratch area was measured using the ImageJ software version 1.25a (NIH, Bethesda, MD, USA) with two randomly selected regions per well.

### 2.6. Tube Formation Assay

In vitro angiogenesis assays were performed using a Matrigel basement membrane matrix (BD Biosciences, Franklin Lakes, NJ, USA; 356234) according to the manufacturer’s instructions. Matrigel was thawed overnight at 4 °C and added to prechilled 48-well plates at 100 μL/well using tips that were pre-chilled and incubated to solidify at 37 °C for 30 min. HUVECs were starved with serum-free ECM for 3 h, digested, collected, and inoculated at 5 × 10^4^ cells per well. miR-21, m/miR-21, or m/NP/miR-21 (25 nM miRNA) were added to the experimental groups, respectively, and PBS was added to the control group. After incubation at 37 °C for 6 h, pictures of tube formation were taken with an inverted microscope (Tengchuan, Ningbo, China; XD-202). The experiment was replicated three times, and the total length of the tubes was measured using the ImageJ software with five randomly selected regions per well.

### 2.7. qRT-PCR

Total RNA from tissue samples (half) and treated cells was extracted with Trizol (Ambion, Austin, TX, USA; 15596-026/018) and quantified using Nanodrop2000 (Thermo Fisher, Waltham, MA, USA). Reverse transcription was performed with a 5× primerscript RT master mix (Vazyme, Suzhou, China; R323-01). Thereafter, qPCR was performed on a Bio-Rad CFX96 PCR system with TB Green Premix Taq (Vazyme, China; Q711-02) and appropriate primers (Table 1; Genewiz, China). The following kits were used for miRNA extraction and quantification: SteadyPure Tissue Cell Small RNA Extraction Kit (AccurateBio, Changsha, China; AG21027); miRNAs 1st Strand cDNA Synthesis Kit (stem loop) (Vazyme, China; MR101); 2x miRNAs Universal SYBR qPCR Master Mix (Vazyme, China; MQ101). 

The housekeeping gene glyceraldehyde-3-phosphate dehydrogenase (GAPDH) was selected as the internal reference gene. Relative levels of mRNAs were calculated using the 2^−ΔΔCT^ method.

### 2.8. Western Blot

Skin tissues and treated cells were lysed with RIPA buffer (Beyotime, China; P0013D) to extract proteins, and proteins were quantified using a BCA protein assay (Tiangen, Beijing, China; pa115-02). Equal amounts (10 µg) of proteins were incubated with the following primary antibodies: anti-VEGFA (Abcam, Cambridge, UK; ab46154), anti-Akt (CST, Danvers, MA, USA; 4691S), anti-pAkt (CST, USA; 4060S), anti-PTEN (CST, USA; 9559), and GAPDH (Abcam, UK; ab9485) were incubated, respectively, overnight at 4 °C and semi-quantified using the Image Pro Plus software version 6.0 (Media Cybernetics, Bethesda, MD, USA).

### 2.9. Mouse Skin Wound Model and Treatment

C57BL/6 male mice (Spfbiotech, Beijing, China) aged 6–8 weeks were used in this study. A two-layer patch method [34] was used for wound preparation. The mice were shaved, disinfected, and coated with a 3 cm × 2 cm antibacterial film (Drape Antimicrobial, Paul, MN, USA; REF6640) before two excisional wounds of 8 mm in diameter were created on their backs. Twenty wounds were randomly divided into 4 groups. Treatments were performed on days 0, 3, and 6 post-operation, and 20 μL (containing 1 μg of miRNA) of miR-21, m/miR-21, m/NP/miR-21, or an equivalent volume of saline was uniformly applied to the wound sites. Then, another 3 cm × 2 cm antibacterial film was used to cover the back skin to maintain sterility and prevent skin contraction. Wound images were acquired on postoperative days 0, 3, 6, 9, 12, and 15 in a vertical view, and the wound area was calculated after calibration with ImageJ. These mice were sacrificed via CO_2_ inhalation on postoperative day 15, and wound specimens (Φ = 8 mm) containing the wound bed and regenerated surrounding skin tissue were harvested. 

### 2.10. Histochemistry and Immunofluorescence

Tissue samples were cut along the midline. Half was used for total RNA or protein extraction; the other half was fixed with 4% paraformaldehyde and then paraffin-sectioned (6 μm thickness). Sections were processed for hematoxylin and eosin (H&E) staining or immunofluorescence (IF) staining using the following primary and secondary antibodies: anti-K14 (Abcam, UK; ab181595), anti-α-SMA (Abcam, UK; ab5694), anti-CD34 (Abcam, UK; ab8158), anti-K15 (SAB, Greenbelt, MD, USA; 48891), anti-SCD (Sigma, St. Louis, MO, USA; HPA012107), donkey anti-rabbit IgG(555) (Abcam, UK; ab150074), and donkey anti-rat IgG(488) (Abcam, UK; ab150153). Fluorescent images were observed and acquired using a confocal microscope (Nikon, Japan; A1R HD25). 

The number of samples (n) was 5 per group in H&E staining, cytokeratin 14 (K14), and CD34/α-smooth muscle actin (CD34/α-SMA) IF staining. Dermal reconstruction was evaluated using the dermal thickness measured from the top edge of the dermis or the epidermal–dermal junction to the dermal–fat junction; 3 measurement points were taken for each image and averaged [35].

Re-epithelialization was evaluated by calculating the epithelial gap (EG) as follows: EG = % (gap length between K14^+^ epidermis/length of tissue). Angiogenesis was measured using the number of CD34^+^/α-SMA^+^ vessel-like structures in the wound center (C) adjacent to the transition zone (T). For IF staining against cytokeratin 15 (K15) and stearoyl-CoA desaturase (SCD), n = 3 per group, and 2 microscopic fields were observed per sample. Images were processed using the ImageJ software.

### 2.11. Statistical Analysis

All results are expressed as the mean ± standard error. Normality was verified using the Shapiro–Wilk test with Brown–Forsythe correction for significantly different SDs. Parametric tests were used to analyze and compare multiple values, with differences being considered statistically significant at *p* < 0.05 (*). Results were plotted using GraphPad Prism 9.0.

## 3. Results

### 3.1. Preparation and Characteristics of Nanocarriers

To investigate whether membrane-coated nanocarriers could act as proper carriers to deliver miRNA, we first tested their basic features. 

The morphologies of the different nanocarriers were visualized using TEM. As displayed in Figure 1A, the mNVs, m/miR-NC, and m/NPs/miR-NC were all observed to be cup- and sphere-shaped in morphology. The mNVs and m/miR-NC unloaded with PLGA-PEI showed vacuoles, whereas m/NP/miR-NC loaded with PLGA-PEI had a dense core and a “dark–light–dark” lipid bilayer surface. Notably, the TEM graphics displayed better homogeneity in the m/NP/miR-NC particles than in the other two groups. The incorporation of PLGA-PEI made the m/NP/miR-NC particles more stable with respect to each other.

The Coomassie Brilliant Blue staining showed that the protein expression profiles of the prepared nanocarriers were consistent with those of cell membranes derived from mADSCs (Figure 1B), indicating that the cell membranes, including the protein components therein, were successfully transferred to the nanocarriers. Co-extrusion enabled the nanocarriers to obtain compositionally complete biomimetic modifications.

The average particle sizes of mNVs, m/miR-NC, and m/NPs/miR-NC were measured to be 167.5 nm, 172.4 nm, and 198.4 nm via DLS (Figure 1C), and the zeta potentials were −22 mV, −19 mV, and −10 mV (Figure 1D). The increased particle size indicated successful loading of PLGA-PEI, and the incorporation of positive PLGA-PEI nanoparticles appeared to reduce the electronegativity of the complexes, which could potentially lead to increased cellular uptake. The NTA data showed that the mean size of m/NP/miR-NC was 111.8 nm in diameter and ranged from 41 to 190 nm (Figure 1E). Meanwhile, m/NP/miR-NC had the highest peak and narrowest peak base among the three groups, indicating that its particle size distribution was more concentrated and had a better degree of homogeneity, which was consistent with the results of TEM.

Stability of the prepared m/miR-NC and m/NP/miR-NC was then determined. As shown in Figure 1F, both m/miR-NC and m/NP/miR-NC were able to maintain a stable particle size for 7 consecutive days, showing good stability with the membrane coating.

The RNA-loading capacity of the two nanocarriers was determined using the encapsulation rate. As shown in Figure 1G, PLGA-PEI incorporation enabled m/NP/miR-NC to load miRNAs more efficiently than m/miR-NC. The results of nucleic acid electrophoresis showed that m/NP/miR-NC was able to release miRNAs consistently and stably in a serum environment for at least 8 h, whereas miRNAs in m/miR-NC were degraded within 2 h (Figure 1H). PLGA-PEI enabled m/NP/miR-NC to load more miRNAs while also providing it with certain slow-release capabilities.

### 3.2. Cellular Uptake Behavior of m/NP/miR-21

To determine the biological effects of m/NP/miR-21 on cells, we first tested the cytotoxicity of the different nanocarriers. 

As shown in Figure 2A, NP/miR-NC-treated cells showed a significant decrease in cell viability compared with the PBS-treated group (PBS). However, cell viability was recovered in the m/miR-NC- and m/NP/miR-NC-treated groups, indicating that biomimetic modifications with mADSC membranes rescued the cytotoxicity of cationic PLGA-PEIs. The treatment of mouse wounds with NP/miR-NC slowed down wound healing, while treatment with m/NP/miR-NC rescued the rate of wound healing (Figure 2B,C).

We next examined the small RNA transport capacity of the different nanocarriers. The uptake of the nanocarriers by different cells was examined. As shown in Figure 2D,E, the PKH26 signal used to label the membrane was significantly enhanced in m/NP/miR-NC-treated cells compared with m/miR-NCs, suggesting that m/NP/miR-NC was more readily taken up. FAM-labeled miR-21s were observed to be aggregated more in m/NP/miR-21-treated hDFs than in m/miR-21-treated ones (Figure 2F). A significant increase in the miR-21 level was also detected via qPCR in hDFs with m/NP/miR-21 treatment in comparison with the m/miR-21 treatment, although the miR-21 levels were already significantly higher in the m/miR-21-treated group than in the miR-21-treated group (Figure 2G).

### 3.3. m/NP/miR-21 Accelerates Mouse Skin Wound Repair

The effects of the nanocarriers on excisional wounds in mice were evaluated according to the experimental paradigm shown in Figure 3A.

m/NP/miR-21-treated mice showed faster wound closure than saline-, miR-21s-, and m/miR-21-treated groups at D3, 6, 9, 12, and 15 (Figure 3B). Wound area was substantially reduced in the m/NP/miR-21-treated group at D6, 9, 12, and 15 compared with that in the miR-21-treated group, and it was also significantly decreased at D6 and D15 in comparison with that in the m/miR-21-treated group (Figure 3C). 

To further explore the effects of the nanocarriers on the wound-healing process, we evaluated wound re-epithelialization, vascularization, and dermal reconstruction in mice after different treatments.

### 3.4. m/NP/miR-21 Promotes Re-Epithelialization by Facilitating Keratinocyte Migration

IF staining against K14, a marker of epithelial cells, revealed that m/NP/miR-21 treatment led to the smallest EG among the four groups (Figure 4A). Wound sites treated with m/miR-21 had a lower gap length ratio than that of the miR-21 group, but nearly all wounds in the m/NP/miR-21 group reached closure (Figure 4B), suggesting that m/NP/miR-21 had a stronger ability to promote re-epithelialization in mouse wounds.

As the major cell type in the epithelium, keratinocytes were employed to detect the effect of m/NP/miR-21 on the promotion of cell migration in vitro (Figure 4C). Compared with the miR-21-treated group, the m/miR-21-treated group had a significantly smaller scratch area after 24 h of treatment. Meanwhile, the m/NP/miR-21 had a remarkably smaller scratch area than that of m/miR-21. Notably, the area of scratches in the m/NP/miR-21-treated Hacats was still significantly smaller than that of the m/miR-21-treated ones 48 h after treatment, by which time the m/miR-21 treatment no longer had an advantage over the miR-21 treatment (Figure 4D). The relative expression levels of genes such as *MMP1*, *MMP3*, and *CDH2*, which promote Hacat migration, were significantly increased after the m/NP/miR-21 treatment (Figure 4E), indicating that PLGA-PEI loading may have given the m/NP/miR-21 a slow-release capability that led to a long-term effect. 

### 3.5. m/NP/miR-21 Accelerates Angiogenesis

Newly formed vessels at wound sites were characterized using IF staining against CD34, a marker of endothelial cells, and α-SMA, a marker of pericytes (Figure 5A). The m/NP/miR-21-treated group had the largest number of mature CD34^+^/α-SMA^+^ vessels at day 15 compared with the miR-21- and m/miR-21-treated groups (Figure 5B). Notable upregulation of VEGFA in wound tissues was observed after treatment with m/NP/miR-21 in comparison with treatment with miR-21 or m/miR-21 (Figure 5C–E). K15, a specific marker of stem cells of the hair follicle bulge, was used for HF labeling, and SCD was used for SGs (Figure 5F). Increased numbers of HFs and SGs in the m/miR-21 and m/NP/miR-21 groups were also observed relative to the saline and miR-21 groups (Figure 5G,H). m/NP/miR-21 promoted the regeneration of skin appendages along with vascularization at the wound site, which improved the functional integrity of regenerated skin.

The regulatory effects of nanocarriers on endothelial cells were also studied in vitro. The tube formation assay showed that both the m/miR-21 and m/NP/miR-21 treatments had effects on the promotion of tube formation at 3 h and 6 h (Figure 5I). The total length of tubular structures treated with m/NP/miR-21 was significantly increased compared with that of the m/miR-21 group at both time points of observation (Figure 5J).

### 3.6. m/NP/miR-21 Promotes Dermal Reconstruction by Enhancing Fibroblast Function

H&E staining showed that the m/NP/miR-21-treated group had the thickest dermis among all four groups (Figure 6A,B), which may have been related to the enhanced function of fibroblasts.

Collagen levels in hDFs, which are the main source of collagen in the skin, were also regulated. The expression of collagen type I alpha 1 chain (COL1A1) was significantly upregulated after the treatment with m/miR-21 in comparison with that of the miR-21-treated mice, while the treatment with m/NP/miR-21 had a more prominent effect than that of the m/miR-21 treatment (Figure 6C–E). Significantly higher levels of COL1A1 were observed in hDFs after the m/NP/miR-21 treatment than after the m/miR-21 treatment (Figure 6H,I).

Proliferation of hDFs was promoted after treatment with the nanocarriers. The expression of Ki66in fibroblasts was significantly increased after the m/miR-21 treatment compared with the miR-21s treatment alone, and the treatment with m/NP/miR-21 led to a further significant enhancement on the basis of m/miR-21 (Figure 6F,G).

Both m/miR-21 and m/NP/miR-21 increased the migration of hDFs at 24 h and 48 h post-treatment (Figure 6J). Compared with m/miR-21-treated hDFs, the scratch area was markedly decreased at 24 h after the m/NP/miR-21 treatment and remained significantly decreased at 48 h after treatment (Figure 6K).

More efficient miR-21 transport resulted in enhanced proliferation, migration, and collagen formation of m/NP/miR-21-treated fibroblasts compared with those treated with m/miR-21, leading to better dermal reconstruction of m/NP/miR-21-treated mouse wounds.

The highest rate of wound closure, the narrowest epithelial gap, the greatest number of mature vessels, and the most newly formed hair follicles and sebaceous glands were observed in the m/NP/miR-21-treated group at D15 after wounding when compared with those of the saline-, miR-21-, and m/miR-21-treated groups, suggesting that m/NP/miR-21 improved the quality of skin wound healing while accelerating the healing process.

### 3.7. m/NP/miR-21 Regulates Skin Repair via the PTEN/Akt Axis

To better understand the potential mechanisms by which exogenous miR-21 regulates skin repair processes, we next employed bioinformatic tools to identify potential target genes of miR-21. We found that the gene PTEN, which regulates angiogenesis and fibroblast function, was identified in different databases. 

We first predicted the binding of miR-21 to PTEN using the online TargetScan tool [36] (Figure 7A). The regulation of PTEN mRNA levels in wound tissue treated with different nanocarriers was then detected, and significant downregulation was revealed in the miR-21-treated group in comparison with the saline-treated group, which indicated the efficient functioning of miR-21 in reducing the mRNA levels of PTEN in mice. The treatment with m/miR-21 was able to further reduce the expression of PTEN, but there were no significant differences. In contrast, the m/NP/miR-21 treatment further drastically reduced the expression levels of PTEN in comparison with those of the m/miR-21 treatment (Figure 7B). This was consistent with the higher miR-21 translocation efficiency of m/NP/miR-21 in the in vitro experiments.

A significant decrease in the PTEN level and an increase in pAkt (S473) and COLI levels were revealed in the m/NP/miR-21-treated wound tissue in comparison with the saline-treated group.

The quantitative Western blotting images and relative band intensity data showed that the PTEN expression level decreased and Akt phosphorylation increased after the miR-21 treatment. The most significant changes were observed after the m/NP/miR-21 treatment (Figure 7C,D).

Inhibitors of Akt phosphorylation significantly inhibited the nanocomplex-induced upregulation of functional genes (Figure 7E,F). Thus, the PTEN/Akt axis may be loaded with potential mechanisms by which miR-21 nanocarriers regulate skin repair.

## 4. Discussion

Cell-membrane-camouflaged nanoparticles are a unique biomimetic nanodrug platform made by wrapping natural cell membranes around nanoparticle cores to mimic the biological structures of original cells. This technology started with erythrocyte-membrane-coated nanoparticles, which replicated the long-circulating characteristics of natural erythrocyte membranes and were attractive for drug transport [20]. Soon, these nanoparticles were used as sensitive erythrocyte decoys to intercept and neutralize pathological agents, such as pore-forming toxins, case antibodies, and chemical toxicants [37,38,39,40]. Meanwhile, red blood cells (RBC)-membrane-encapsulated nanoparticles retaining bacterial toxins were used as nanotoxins to trigger a protective immune response against bacterial infection [41]. After initial development, membranes of other cell types were derived for surface encapsulation. The resulting nanoparticles enabled new therapeutic opportunities through cell mimicry and multifaceted biointerfacial properties.

In this study, we constructed a miRNA nanocarrier with a high encapsulation rate and efficient translocation by mimicking the original shell–nucleus structure of cells. Extrusion of the polycarbonate membrane with a fixed pore size brought it to a homogeneous particle size of about 190 nm and successfully preserved the protein components on the cell membrane.

In order to better enrich microRNAs during the preparation process, we used PLGA-PEI nanoparticles as the core and loaded them with microRNAs through electrostatic adsorption [17]. The miR-21 loading efficiency was successfully increased from ~50% to ~95% by using PLGA-PEI as the adsorptive core. However, cationic organic nanoparticles usually have a certain cytotoxicity [14,18,19], while stem-cell-derived cell membranes inherit the low immunogenicity of MSCs [42,43], which can effectively reduce the cytotoxicity of nanoparticles [44]. In line with the results of these studies, the cytotoxicity of the prepared m/NP/miR-21 was significantly reduced when we encapsulated mADSC membranes with NP/miR-21. In addition, Patil et al. [45] showed that cellular uptake decreased from 1.55 μg/L to 0.25 μg/L when the nanoparticle surface potential was increased from −43 mV to −16 mV. Consistent with this, m/NP/miR-21 was able to translocate more miR-21 into the cells. On the one hand, this was due to the fact that PLGA-PEI was enriched with more miR-21. On the other hand, it was due to the cations carried by PLGA-PEI making the complex less electronegative and more easily taken up by the cells [46,47].

Angiogenesis, the formation of new blood vessels from the pre-existing vascular system, is another important process in wound healing [48]. As endothelial cells degrade the surrounding extracellular matrix, they begin to migrate, proliferate, and interconnect to form new blood vessels [49]. The blood vessels, along with macrophages and fibroblasts, move into the wound space as a single unit and are replenished with nutrients and oxygen to maintain cellular metabolism [50]. In addition, fibroblasts can assist in the process through extracellular matrix remodeling and local delivery of growth factors [51]. Exosomes from MSCs have been reported to be internalized by HUVECs and to promote endothelial angiogenesis during wound repair [52,53]. In contrast to m/miR-21, we found that m/NP/miR-21 was similarly more readily internalized by HUVECs in vitro. In vitro, m/NP/miR-21 further promoted tube formation and VEGFA expression in HUVECs. All of these effects of m/NP/miR-21 were shown to enhance wound healing, with higher rates of wound closure and more neointimal formation in mice.

Fibroblasts, from the late inflammatory stage until complete epithelialization, are crucial in the wound-healing process [54,55]. They are called to migrate to the wound area and proliferate to participate in wound contraction, extracellular matrix deposition, tissue remodeling, etc. [56,57]. Previous studies have demonstrated that exosomes derived from MSCs can be internalized to regulate fibroblast proliferation and migration [58,59]. Meanwhile, in this study, compared with m/miR-21, m/NP/miR-21 was more favorable for entering hDFs in vitro and promoting cell migration, proliferation, and pro-fibrotic gene expression. In vivo experiments further confirmed that all of these effects of m/NP/miR-21 would promote wound healing in mice with higher wound closure rates and fibroblast activation.

PTEN are dual phosphatases and can antagonize phosphatidylinositol-4,5-bisphosphate 3-kinase (PI3K) activity by converting PI (3,4,5) P3 into PI (4,5) P2, which plays an important role in phosphorylating Akt to achieve the negative regulation of the Akt/PI3K signaling pathway [60]. The activated PI3K/Akt pathway not only upregulates VEGF expression but also promotes cell proliferation, migration, angiogenesis, and collagen synthesis and stimulates wound healing [61].

miR-21 is generally considered to be a microRNA that regulates the wound-healing process through the PTEN/Akt signaling pathway [62]. Yan et al. [63] reduced miR-21-5p expression to inhibit autophagy and increase apoptosis in keloid fibroblasts and found that miR-21-5p regulated migration and autophagy-related gene expression through PTEN and pAkt signaling. These studies suggest that miR-21 is involved in wound healing by inhibiting PTEN’s inactivation of the PI3K/Akt signaling pathway. Similar results were also validated in this study. The mRNA and protein levels of PTEN were found to be downregulated after miR-21 treatment, suggesting that miR-21 can target and inhibit the expression of PTEN, and PTEN was downregulated more after m/NP/miR-21 treatment. pAkt levels were increased after the downregulation of PTEN, but no significant changes were observed in the Akt protein levels, suggesting that the PTEN gene has a negative regulatory effect on pAkt. Activated pAkt significantly promotes the activation and proliferation of dermal fibroblasts, as well as the reconstitution of the extracellular matrix, the release of vascular endothelial growth factor, and the upregulation of the expression of functional proteins, including VEGFA and COL1 [64,65]. Similar results were also seen in our study, and the upregulation of expression was most pronounced after m/NP/miR-21 treatment. miR-21’s promotional effect on blood vessel formation and collagen reconstruction was eliminated by the Akt inhibitor. Taken together, as miRNA carriers, cell-membrane-coated PLGA-PEI nanocarriers are capable of efficient microRNA transport to a wide range of cells, and PLGA-PEI loading enhances the cellular uptake of miR-21. The miR-21 transported into cells by the nanocarriers mainly regulates the Akt phosphorylation level by targeting PTEN and activating the PI3K/Akt signaling pathway to accelerate the skin-healing process. This provides a relatively safe and efficient methodological reference for the in vivo delivery of miRNA.

## 5. Conclusions

In summary, we developed a miRNA delivery system for stem-cell-membrane-encapsulated exosomes mimicking nanocomplexes for repairing skin damage. PLGA-PEI nanoparticles were found to achieve high-capacity miRNA loading and effective protection of miRNA, and the biomimetic modification brought by stem cell membranes reduced the cytotoxicity of the PLGA nanoparticles, making the nanocarriers have better biocompatibility. These nanocarriers can be taken up by a variety of cells derived from the skin, which provides miR-21 with a wider range of targets at the wound site and fully regulates a variety of biological processes involved in wound repair. The released miR-21 silenced PTEN mRNA and promoted fibroblast migration, proliferation, and collagen synthesis while promoting keratinocyte migration and endothelial cell tubule formation. In animal models of skin defects, nanocarriers significantly accelerated the skin-repair process and reinstated the skin function at the wound site by promoting re-epithelialization, dermal reconstruction, and revascularization. In conclusion, our exosome-mimetic nanocomposite-based miRNA delivery system offers high biocompatibility and transfection efficiency, and it has promising prospects for the treatment of skin lesions and miRNA applications.

## Figures and Tables

**Figure 1 pharmaceutics-16-01113-f001:**
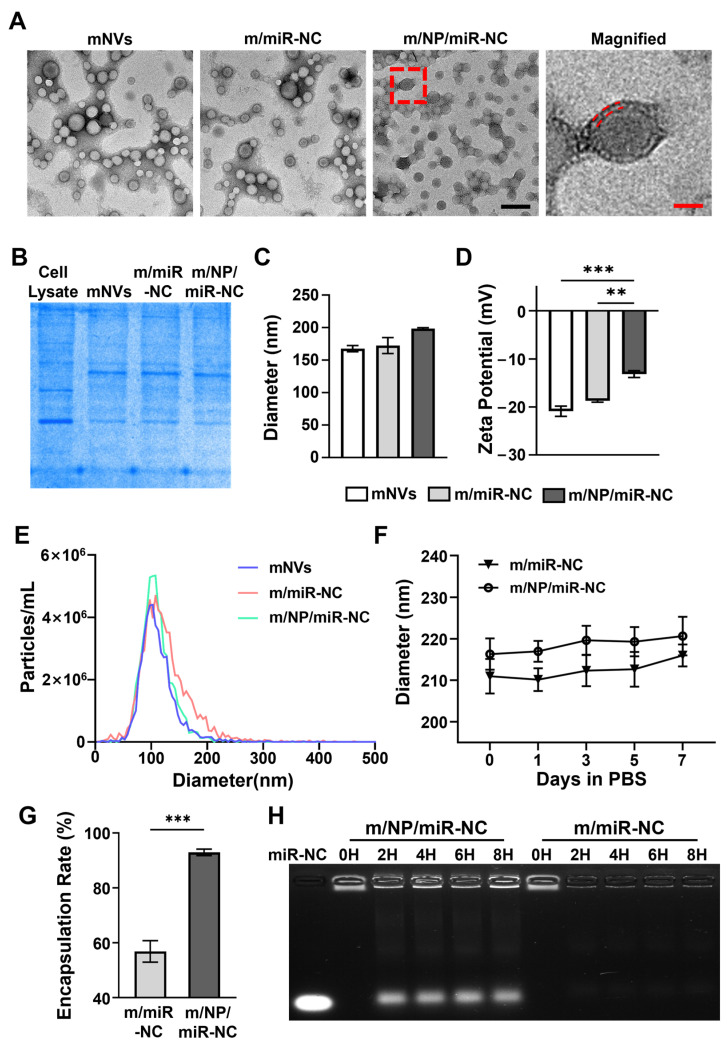
Preparation and characterization of biomimetic nanocarriers. (**A**) Surface morphology of mNVs, m/miR-NC, and m/NP/miR-NC detected by TEM, where magnification of the image in the red dashed box is shown on the right, the red dashed lines in Magnified represent the bilayer structure of the cell membrane, scale bar (black) = 200 nm, and scale bar (red) = 40 nm; (**B**) Coomassie Brilliant Blue staining expression profile of cell membrane proteins in cell lysate, mNVs, m/miR-NC, and m/NP/miR-NC; (**C**) DLS detection of the surface potential of mNVs, m/miR-NC, and m/NP/miR-NC; (**D**) DLS detection of the average particle size of mNVs, m/miR-NC, and m/NP/miR-NC; (**E**) NTA detection of the particle size distribution of mNVs, m/miR-NC, and m/NP/miR-NC; (**F**) DLS detection of stability of m/miR-NC and m/NP/miR-NC; (**G**) miRNAs encapsulation rate of m/miR-NC and m/NP/miR-NC; (**H**) Nucleic acid electrophoresis to detect the degradation of miRNAs in m/miR-NC and m/NP/miR-NC in 10% FBS. ** *p* < 0.01, *** *p* < 0.001.

**Figure 2 pharmaceutics-16-01113-f002:**
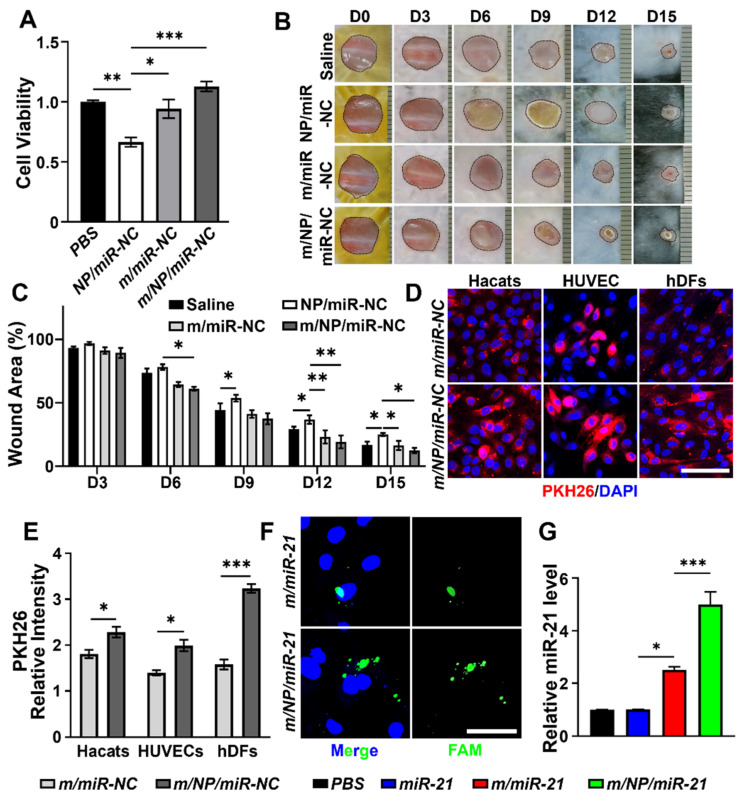
Cellular uptake behaviors of biomimetic nanocarriers. (**A**) Cytotoxicity of NP/miR-NC, m/miR-NC, and m/NP/miR-NC detected by CCK8, n = 5; (**B**) Images of dynamic changes in saline-, NP/miR-NC-, m/miR-NC-, and m/NP/miR-NC-treated wounds; (**C**) Statistical analyses of wound area, n = 5; (**D**) Fluorescence staining results showed that cells such as Hacats, HUVECs, and hDFs all had higher uptake of m/NP/miR-NC compared to m/miR-NC, scale bar = 50 nm; (**E**) Statistical results of cellular uptake immunofluorescence plots, n = 5; (**F**) Fluorescence staining showed that m/NP/miR-21 translocated more miR-21 into the adult fibroblasts, scale bar = 20 μm; (**G**) Relative expression of miR-21 in fibroblasts after PBS, miR-21, m/miR-21, and m/NP/miR-21 treatment, n = 9. * *p* < 0.05, ** *p* < 0.01, *** *p* < 0.001.

**Figure 3 pharmaceutics-16-01113-f003:**
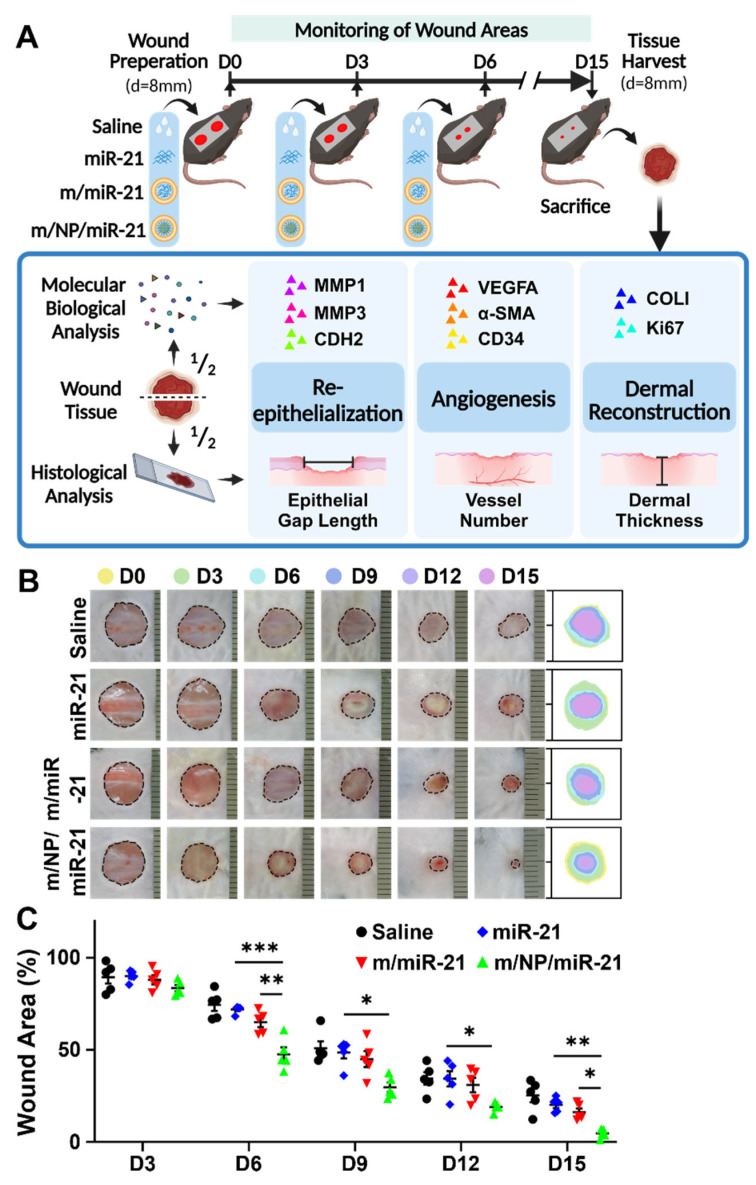
m/NP/miR-21 accelerates skin wound healing in mice. (**A**) Schematic diagram of in vivo experiment; (**B**) Images of dynamic changes of the wound after saline, miR-21, m/miR-21, or m/NP/miR-21 treatment; (**C**) Statistical analysis of the wound area, n = 5. * *p* < 0.05, ** *p* < 0.01, *** *p* < 0.001.

**Figure 4 pharmaceutics-16-01113-f004:**
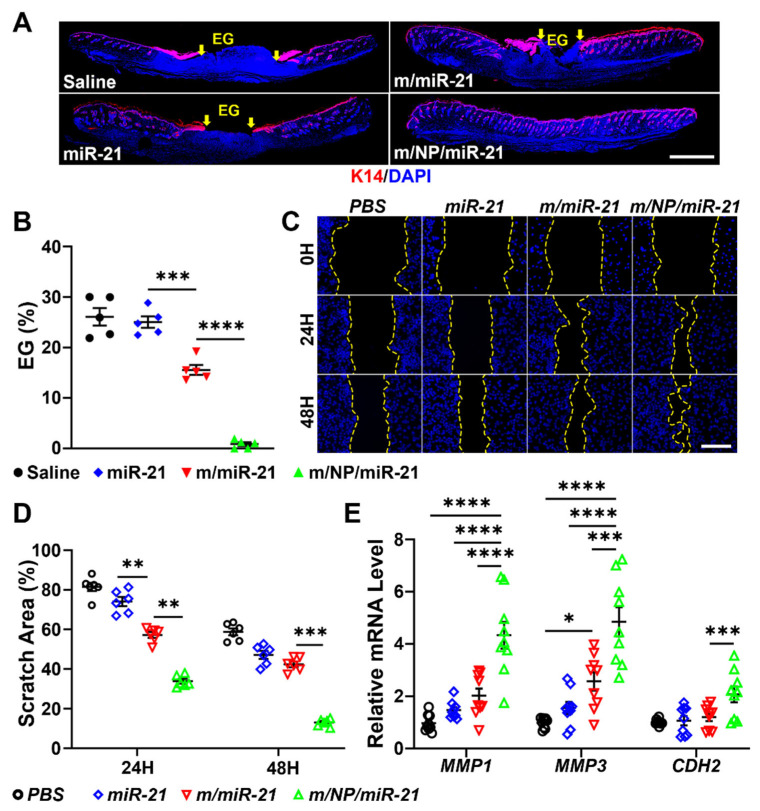
m/NP/miR-21 accelerates re-epithelialization. (**A**) Images of K14 immunofluorescence staining of saline, miR-21, m/miR-21, or m/NP/miR-21 treatment groups, between the yellow arrows is the epithelial gap (EG), scale bar = 1 mm; (**B**) Statistical analysis of EG, n = 5; (**C**) Images of dynamic changes of Hacats scratches after PBS, miR-21, m/miR-21, or m/NP/miR-21 treatment, scale bar **=** 200 μm; (**D**) Statistical analysis of scratch area, n = 6; (**E**) qPCR detection of genes related to keratocyte migration after saline, miR-21, m/miR-21, or m/NP/miR-21 treatment relative mRNA levels of these genes, n = 9. * *p* < 0.05, ** *p* < 0.01, *** *p* < 0.001, **** *p* < 0.0001.

**Figure 5 pharmaceutics-16-01113-f005:**
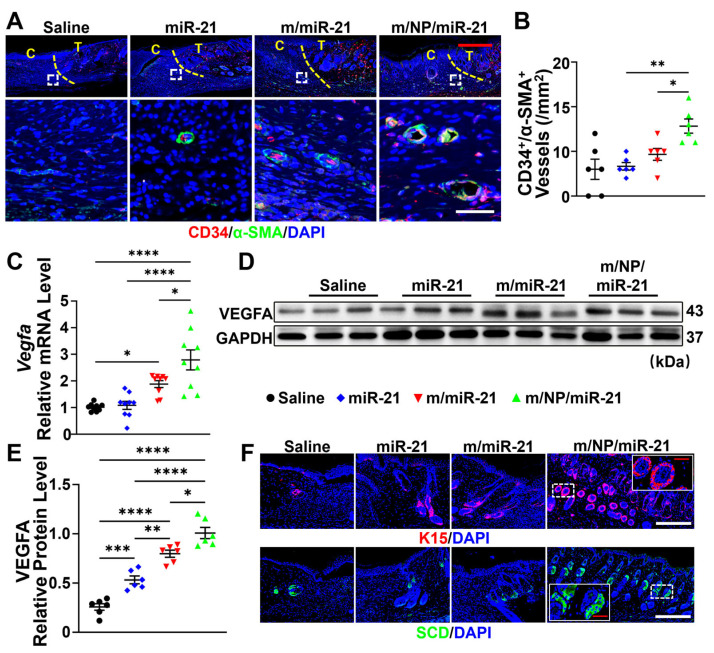
m/NP/miR-21 promotes angiogenesis and skin appendage formation. (**A**) Images of double-labeled immunofluorescence staining for CD34, α-SMA in saline-, miR-21-, m/miR-21-, or m/NP/miR-21-treated groups, the yellow dashed line shows the boundary between the transitional area (T) and the center area (C), images in white dashed squares are shown below, scale bar (white) =50 μm and scale bar (red) = 200 μm; (**B**) Statistical analysis of vascular number of CD34^+^/α-SMA^+^, left and right sides of each tissue were counted as two observation points, n = 6; (**C**) qPCR detection of saline, miR-21, m/miR-21, or m/NP/miR-21 treatment after mRNA expression changes of VEGFA in tissues, n = 9; (**D**) WB detection of protein expression changes of VEGFA in wound tissues after saline, miR-21, m/miR-21, or m/NP/miR-21 treatment; (**E**) semi-quantification of VEGFA protein levels in wound tissues, n = 6; (**F**) Images of immunofluorescence staining of K15 or SCD labeling in saline-, miR-21-, m/miR-21-, or m/NP/miR-21-treated groups with scale bar (white) = 100 μm and scale bar (red) = 200 μm; (**G**) Statistical analysis of the number of SCD^+^ sebaceous glands (SGs), left and right sides of each tissue were counted as two observation points, n = 6; (**H**) Statistical analysis of the number of K15^+^ hair follicles (HFs), left and right sides of each tissue were counted as two observation points, n = 6; (**I**) Images of dynamic changes in tubule formation of HUVECs after PBS, miR-21, m/miR-21, or m/NP/miR-21 treatment, scale bar = 200 μm; (**J**) Statistical analysis of the total length of tube formation, n = 3. * *p* < 0.05, ** *p* < 0.01, *** *p* < 0.001, **** *p* < 0.0001.

**Figure 6 pharmaceutics-16-01113-f006:**
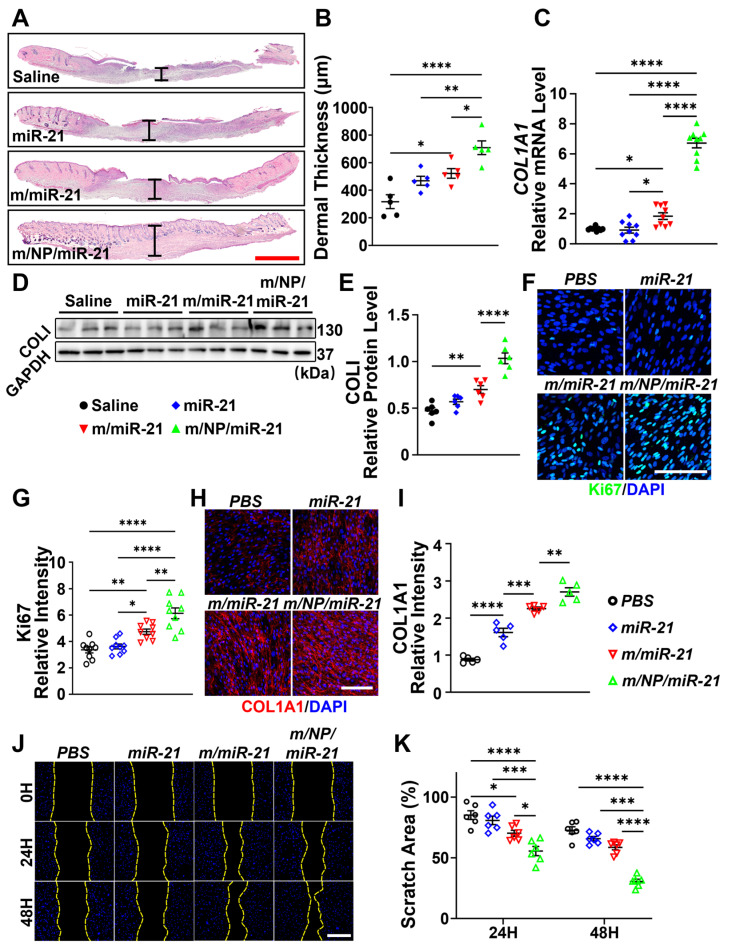
m/NP/miR-21 regulates collagen formation to promote dermal reconstruction. (**A**) H&E staining to detect dermal thickness, measured from the top edge of tissue or epidermal–dermal junction to the dermal–fat junction, changes in wound tissues after saline, miR-21, m/miR-21, or m/NP/miR-21 treatments, scale bar = 1 mm, with I-shaped black line indicating the measurement of the dermal thickness; (**B**) Statistical analysis of dermal thickness, n = 5; (**C**) qPCR detection of saline-, miR-21-, m/miR-21-, or m/NP/miR-21-treated wound mRNA expression changes of type I collagen (COLI) in tissues, n = 9; (**D**) WB detection of protein expression changes of COLI in wound tissues after saline, miR-21, m/miR-21, or m/NP/miR-21 treatment; (**E**) Semi-quantification of COLI protein levels in wound tissues, n = 6; (**F**) Immunofluorescence staining plots of Ki67 labeling in PBS, miR-21, m/miR-21, or m/NP/miR-21 treatment, scale bar = 200 μm; (**G**) Statistical analysis of the fluorescence intensity of Ki67, n = 9; (**H**) Immunofluorescence staining of COL1A1 in PBS, miR-21, m/miR-21, or m/NP/miR-21 treatment hDFs, scale bar =200 μm; (**I**) Statistical analysis of fluorescence intensity of COL1A1, n = 5; (**J**) Images of dynamic changes of hDFs scratches after PBS, miR-21, m/miR-21, or m/NP/miR-21 treatment, scale bar = 200 μm; (**K**) Statistical analysis of scratch area, n = 6. * *p* < 0.05, ** *p* < 0.01, *** *p* < 0.001, **** *p* < 0.0001.

**Figure 7 pharmaceutics-16-01113-f007:**
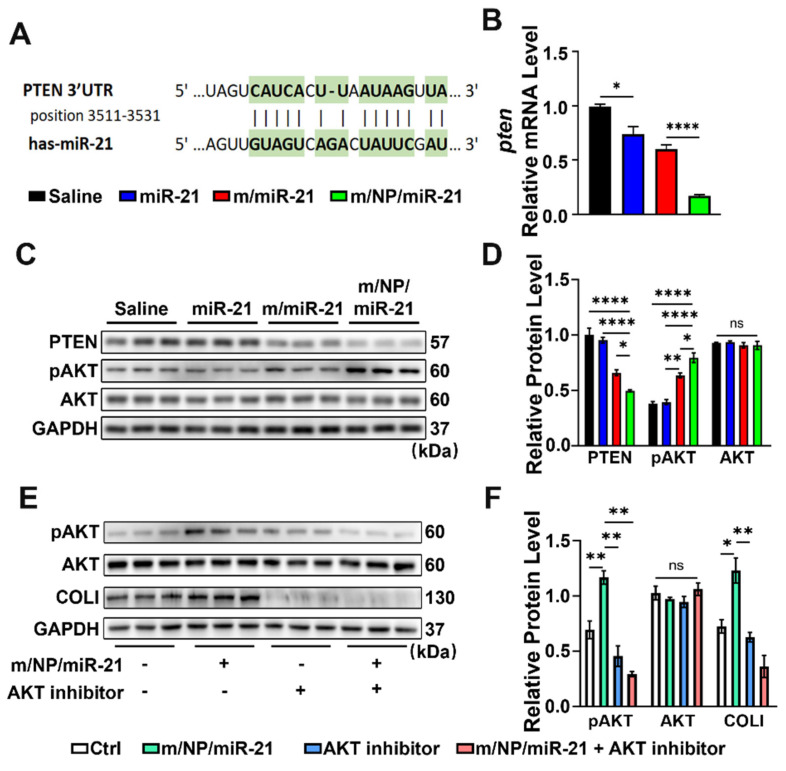
m/NP/miR-21 regulates skin wound healing via PTEN/Akt pathway. (**A**) TargetScan predicts that miR-21 binds to the 3′-UTR region of PTEN, with green highlighting and | indicating the binding sites; (**B**) qPCR detects mRNA expression changes of PTEN in wound tissues after saline, miR-21, m/miR-21, or m/NP/miR-21 treatment, n = 9; (**C**) WB detection of PTEN protein expression changes of PTEN, AKT, and pAKT in wound tissues after saline, miR-21, m/miR-21, or m/NP/miR-21 treatments; (**D**) Semiquantitative analysis of protein expression of PTEN, AKT, and pAKT, n = 6; (**E**) WB detection of inhibition of AKT activation; (**F**) Semiquantitative analysis of inhibition of AKT, n = 6. ns *p* > 0.05, * *p* < 0.05, ** *p* < 0.01, **** *p* < 0.0001.

**Table 1 pharmaceutics-16-01113-t001:** Primer sequences for qRT-PCR.

Gene Names	Primer Sequences
*MMP1*	forward: 5′-ATGAAGCAGCCCAGATGTGGAG-3′
	reverse: 5′-TGGTCCACATCTGCTCTTGGCA-3′
*MMP3*	forward: 5′-CACTCACAGACCTGACTCGGTT-3′
	reverse: 5′-AAGCAGGATCACAGTTGGCTGG-3′
*CDH2*	forward: 5′-CGAAAACAGGCACTGGACCACT-3′
	reverse: 5′-GATGACGACTGTGTCCGCTGAA-3′
*COL1A1*	forward: 5′-GATTCCCTGGACCTAAAGGTGC-3′
	reverse: 5′-AGCCTCTCCATCTTTGCCAGCA-3′
*GAPDH*	forward: 5′-GTCTCCTCTGACTTCAACAGCG-3′
	reverse: 5′-ACCACCCTGTTGCTGTAGCCAA-3′
*Vegfa*	forward: 5′-CTGCTGTAACGATGAAGCCCTG-3′
	reverse: 5′-GCTGTAGGAAGCTCATCTCTCC-3′
*pten*	forward: 5′-TGAGTTCCCTCAGCCGTTACCT-3′
	reverse: 5′-GAGGTTTCCTCTGGTCCTGGTA-3′
U6	forward: 5′-CTCGCTTCGGCAGCACAT-3′
	reverse: 5′-TTTGCGTGTCATCCTTGCG-3′

## Data Availability

The data that support the findings of this study are available from the corresponding author upon reasonable request.

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
