# Peer review of "Adipose-Derived Stem-Cell-Membrane-Coated PLGA-PEI Nanoparticles Promote Wound Healing via Efficient Delivery of miR-21"

_pharmaceutics, 2024, doi:10.3390/pharmaceutics16091113_

Round 1
Reviewer 1 Report
Comments and Suggestions for Authors
In the proposed manuscript, the authors present a novel combination of membrane-coated nanoparticles that bind and transfer miRNAs to cells and tissues. The authors aim to develop an exosome-mimetic nanocomposite-based miRNA delivery system with high biocompatibility and transfection efficiency. The study is based on the knowledge of the beneficial effects of MSC in improving wound healing and their membranes in delivering nanoparticles that stabilise and protect transferred RNA molecules. The experiments are well chosen to show the successful development of the biomaterial and to test its biological activity.
However, the main concern is the controls used: The experiments with miR-21 were controlled by PBS treatment. This is not an appropriate control. As is common in the literature, both miR and siRNA experiments need to be controlled by the use of non-coding or scrambled control nucleotides, as the addition of foreign nucleotides can have biological effects. This was also shown by the authors in Figure 2B,C when miRNA mimics were used in their biomaterials and were different from PBS control (Fig.2b,C d6pw). It remains to be shown whether the effects in Figures 3-6 are related to the transferred miR-21 or to the biomaterial components.
The beneficial role of miR-21 in wound healing has been published previously. Therefore, the data shown in Figure 7 are confirmatory. Furthermore, some previous publications should be acknowledged in the discussion section (see https://doi.org/10.1016/j.abb.2021.108895; doi: 10.1016/j.biocel.2019.105570; review: doi: 10.3389/fphar.2022.828627).
In general, the manuscript is difficult to read because of the many abbreviations not introduced by the authors. Therefore, an "Abbreviations" section should be added.
Furthermore, there are several other points which should be addressed:
1. L61-62: please explain this statement more in detail, or omit it.
2. Fig.1 all abbreviations should be explained: mNV, m/miR-NC, m/NPs/mir-NC, NTA and many more. In 1G, the groups should be shown on the x-Axis.
1H: why there is no miRNA at 0h when you state that RNA is lost w/o NP after 2h?
3. L224: if you created round wounds by punching, the epidermal gap/wound size depends on the distance from the middle of the wound the tissue was sliced and shown. How did you ensure to compare tissue specimens at the same distance from wound center?
4. L287: Fig.2A shows number of viable cells, not activity.
5. L291: This sentence better belongs to Fig. 2A
6. Fig.3A suggests that the wounds were supplied with soluble formulations at days 0,3 and 6 post wounding. This fact is not mentioned in the methods section. Please describe more precisely there.
7. Fig.5: The size of the immunofluorescence pictures does not allow to see double-positivity for CD34 an aSMA. This would be important in order to distinguish vessels from aSMA-positive myofibroblasts.
8. Fig.6A: dermal thickness measurement strongly depends on the angle the tissue is cut. How did you ensure that the tissue was cut with the same angle for all specimens? It is not clear from the photos whether the dermis was measured. It looks more like whole wound tissue. I would suggest doing a Masson Goldner trichrome stain and showing blue collagen staining.
9. Discussion of the wound model: These wounds close mainly by contraction. What is known on the effect of miR21 on muscle cells? This fact should be discussed here.
Comments on the Quality of English LanguageEnglish is mostly OK. Some typos are found. Please check again.
Reviewer 2 Report
Comments and Suggestions for Authors
Peng and colleagues investigated the effect of miR-21 on wound healing through in vitro and in vivo studies. To effectively deliver miR-21 into cells and protect it, they encapsulated it into vesicles obtained from mesenchymal stem cell membranes or nanoparticles, which were then covered with MSC membranes.
The article is interesting, and the experiments are appropriately designed. The results collected support the study's conclusions and are adequately discussed.
I only have the following observations:
Section 2.2: please specify how the ratio 1:1 is expressed. Is it a w:w ratio?
Define mNVs, m/miR-NC and m/NPs/miR-NC. It is hard to follow the first part of the results, as those names are very similar, and a complete distinction among the three different samples becomes clear only later.
It is not described how the release studies plotted in Fig 1H have been performed.
Comments on the Quality of English LanguageNone
Round 2
Reviewer 1 Report
Comments and Suggestions for Authors
In the revised version the authors explained their intention of using PBS as negative control in their experiments. The cited several papers doing similar. Although I am not completely convinced in such a setting, previous data suggest that in most cases there is no non-specific effect of the foreign miR if encapsulated by membrane vesicles. Thus, the actual setting might be acceptable.
The paper was checked and the discussion was improved in some aspects. 3 points should be checked:
1. Discussion lane 512/513: There is a duplicate statement here. Please check and remove one sentence.
2. Discussion lane 529/530: Where did you show “appropriate collagen ratios” (how are these defined?) and no risk of scar formation?
This is an over-interpretation of your data shown. No data on early wound collagen type III are shown and there are no findings on scarring.
3. Discussion L541: The sentence "These studies suggest that miR-21 is involved in wound healing by inhibiting PTEN’s activation of the PI3K/Akt signaling pathway. " is incorrect because PTEN would INACTIVATE the PI3K/Akt pathway as mentioned above.
